# CHIVAX 2.1-Loaded Chitosan Nanoparticles as Intranasal Vaccine Candidates for COVID-19: Development and Murine Safety Assessment

**DOI:** 10.3390/biomedicines13102453

**Published:** 2025-10-09

**Authors:** Lineth Juliana Vega-Rojas, Monserrat Palomino, Iván Corona-Guerrero, Miguel Ángel Ramos-López, María Antonieta Carbajo-Mata, Diana Vázquez-Olguín, Juan Campos-Guillen, Aldo Amaro-Reyes, Zaida Urbán-Morlán, José Alberto Rodríguez-Morales, Juan Mosqueda, Héctor Pool

**Affiliations:** 1Immunology and Vaccines Laboratory, Faculty of Natural Sciences, Autonomous University of Querétaro, Campus Aeropuerto, Carretera a Chichimequillas, Ejido Bolanos, Querétaro 76140, Mexico; lineth.vega@uaq.mx (L.J.V.-R.); ivan.corona@uaq.mx (I.C.-G.); dvazquez33@alumnos.uaq.mx (D.V.-O.); 2Secretaria de Ciencias, Humanidades y Tecnologías (SECIHTI), Av. Insurgentes Sur 1582, Alcaldía Benito Juárez, Crédito Constructor, Ciudad de México 03940, Mexico; 3Faculty of Health Sciences, University of the Valley of Mexico (UVM), Naranjos Punta Juriquilla 1000, Santa Rosa Jáuregui, Querétaro 76230, Mexico; A110183385@my.uvm.edu.mx; 4Faculty of Chemistry, Autonomous University of Querétaro, Querétaro 76010, Mexico; miguel.angel.ramos@uaq.mx (M.Á.R.-L.); juan.campos@uaq.mx (J.C.-G.); aldo.amaro@uaq.edu.mx (A.A.-R.); 5Laboratorio Universitario del Bioterio, Instituto de Neurobiología, UNAM, Universidad Nacional Autónoma de México, Querétaro 76230, Mexico; mariacarbajomata@comunidad.unam.mx; 6Center of Drug Information and Clinical Pharmacy, Faculty of Chemistry, Autonomous University of Yucatán, Mérida 97069, Mexico; zaida.urban@correo.uady.mx; 7Research and Graduate Studies Division, Faculty of Engineering, Autonomous University of Querétaro, Querétaro 76010, Mexico; jose.alberto.rodriguez@uaq.mx

**Keywords:** chimeric protein, chitosan nanoparticles, intranasal vaccines, COVID-19, biosafety

## Abstract

**Background/Objectives:** Innovative intranasal delivery systems have emerged as a strategy to overcome the limitations of conventional COVID-19 vaccines, including suboptimal mucosal immunity, limited antigen retention, and vaccine hesitancy. This study aimed to evaluate physicochemical properties and murine safety of a novel COVID-19 intranasal vaccine candidate based on CHIVAX 2.1 (CVX)-loaded chitosan nanoparticles (CNPs). **Methods:** The CVX recombinant protein was encapsulated into CNPs using the ionic gelation method. The nanoparticles were characterized by their physicochemical properties (mean size, zeta potential, morphology, and encapsulation efficiency) and spectroscopic profiles. Mucin adsorption and in vitro release profiles in simulated nasal fluid were also assessed. In vivo compatibility was evaluated through histopathological analysis of tissues in male C-57BL/6J mice following intranasal administration. **Results:** CNPs exhibited controlled size distribution (38.5–542.5 nm) and high encapsulation efficiency (65.4–92.2%). Zeta potential values supported colloidal stability. TEM analysis confirmed spherical morphology and successful CVX encapsulation, and immunogenic integrity was also demonstrated. Mucin adsorption analysis demonstrated effective nasal retention, particularly in particles ≈90 nm. In vitro release studies revealed a biphasic protein profile, where ≈80% of the recombinant protein was released within 2 h. Importantly, histopathological analyses and weight monitoring of intranasally immunized mice revealed no signs of adverse effects related to toxicity. **Conclusions:** The ionic gelation encapsulation process preserved the physical and immunological integrity of CVX antigen. Furthermore, the intranasal administration of the CVX-loaded CNPs demonstrated a favorable safety profile in vivo. These findings support the potential of the CVX intranasal vaccine formulation for further immunogenicity studies, with no apparent biosafety concerns.

## 1. Introduction

Coronavirus disease 2019 (COVID-19), caused by the severe acute respiratory syndrome coronavirus 2 (SARS-CoV-2), presents with a broad spectrum of clinical manifestations ranging from mild symptoms—such as fever, cough, fatigue, headache, sore throat, anosmia, ageusia, and gastrointestinal disturbances—to severe respiratory complications and death [1]. Vaccination has been a cornerstone in mitigating the burden of the disease, significantly reducing the incidence of severe illness and mortality [2]. However, vaccines administered via conventional parenteral routes have shown limited efficacy in preventing viral infection and transmission. This limitation highlights the need for alternative immunization strategies capable of inducing robust mucosal immunity at the site of viral entry. Intranasal vaccine administration provides a more effective immune response in the nasal mucosa—the first point of contact with the virus—and has other advantages over conventional administration [3], including a less invasive route of administration, increased patient acceptance, wider coverage, and the ease of administration in emergency situations [4]. However, administering antigens through the nose can be challenged by the dynamic nature of the nasal mucosa, the presence of defense mechanisms that can degrade antigens, and the fact that antigens are eliminated by mucociliary flow, or “washing,” due to their low adherence to the nasal mucosa. This reduces their exposure time to immune cells, resulting in inefficient activation [5]. To overcome these challenges, substances called adjuvants are added to vaccines to improve antigen adherence to the nasal mucosa, enhancing the immune response and improving vaccination efficacy. Examples of these substances include aluminum, squalene, polysorbate 80 (P80), cetyltrimethyl ammonium bromide (CTAB), and monophosphoryl A (MPLA). They have been implemented to enhance humoral and cellular immune responses [6]; however, adverse effects from their use have been reported, including pain and redness at the injection site, swelling, severe allergic reactions, release of proinflammatory cytokines and histamine, and mast cell activation [7]. The design of polymeric nanoparticles has emerged as an ideal tool for avoiding the negative effects of conventional adjuvants. They can improve the stability of encapsulated antigens, increase their absorption and penetration, and reduce toxicity and inflammation, controlling the antigen release, which enhances the duration of the immune response [8,9]. One of the most widely used materials to create nanoparticles as an antigen carrier adjuvant is chitosan (C), a natural polymer derived from chitin, with excellent adhesive properties for the nasal mucosa, and stimulating immune responses (humoral and cellular) [10]. Due to its biocompatible and biodegradable characteristics, it reduces the risk of adverse reactions and allows elimination from the body [8]. One of the most important aspects to evaluate in new vaccines against SARS-CoV-2 is the biosafety of the formulation. Demonstrating the safety of intranasal vaccine candidates can positively impact public acceptance of newly created formulations against this disease [11]. Therefore, the present study focuses on the evaluation of the characterization of physicochemical properties and murine safety of an intranasal vaccine candidate based on CHIVAX 2.1 (CVX)-loaded chitosan nanoparticles (CNPs). CHIVAX is a chimeric recombinant protein that was developed and patented (MX/A/2021/012438) by members of our working group. The first version of CHIVAX has demonstrated that it can induce protective antibodies (IgG) against SARS-CoV-2 in a porcine animal model when administered subcutaneously or intranasally [12]. Part of our work group has developed another version of CHIVAX called CHIVAX 2.1 (CVX), which was tested in a mouse model with a two-dose administration scheme (0 and 21 days). CVX demonstrated the activation of humoral and cellular responses through specific antibodies (IgG, IgG_2a_, and IgG_2b_), as well as cytokines, such as TNF-α and IFN-γ, and specific CD4+ and CD8+ T cells [13]. This study constitutes the first attempt to encapsulate the recombinant CVX protein in chitosan-based nanoparticles optimized for intranasal delivery and to investigate its physicochemical properties and safety in a murine model.

## 2. Materials and Methods

### 2.1. Materials

Chitosan (C, Wall material), poly (vinyl alcohol) (PVA), bovine serum albumin (BSA), and sodium tripolyphosphate (TPP) were purchased from Sigma-Aldrich (St. Louis, MO, USA). Hydrochloric acid (HCl), sodium hydroxide (NaOH), acetic acid, ethanol (EtOH), sodium chloride (NaCl), monobasic potassium phosphate, dibasic potassium phosphate, and potassium chloride (KCl) were purchased from Karal S.A de C.V. (Guanajuato, Mexico). TBST solution (20 mM Tris-base, 150 mM NaCl, 0.1% Tween-20), transfer buffer (20 mM Tris-base, 120 mM glycine, and 20% methanol) were purchased from GoldBio (St. Louis, MO, USA).

### 2.2. Synthesis of CVX-Loaded C Nanoparticles

The CVX recombinant protein was obtained following the protocols previously described by Vega-Rojas et al. [13] and Mosqueda et al. [12]. Briefly, the multi-epitope recombinant protein was designed to include conserved B- and T-cell epitopes of the SARS-CoV-2 receptor-binding domain. The synthesis involved bioinformatic selection of immunogenic peptides, cloning into expression vectors, and heterologous expression in *E. coli*. Subsequent purification was performed using affinity chromatography under denaturing conditions, ensuring high purity and yield suitable for immunological studies.

The encapsulation of CVX into CNPSs was performed by modifications to the protocols established by Behnaz et al. [14] and Akerele et al. [15]. Purified CVX was dispersed in a Tween-20-enriched PBS (pH 5.5) and added dropwise to a tripolyphosphate (TPP) solution (2 mg/mL, in distilled water) and kept under magnetic stirring (500 rpm) for 60 min at room temperature. This CVX-TPP solution was added gradually to a solution of C at different concentrations (1, 2, and 5 mg/mL, in 5% acetic solution) previously adjusted at pH 5.5. The final mixture (CVX, TPP, and C) was maintained under magnetic stirring (500 rpm) until a cloudy solution was observed, and then maintained for 2 more h, at the same conditions. CVX-CNPSs were collected by ultracentrifugation (23,000 rpm/20 min/8 °C), and the pellet was rinsed before being frozen for 24 h (−20 °C) and finally lyophilized. Lyophilized nanoparticles were stored in sterile amber vials in humidity-free desiccators at 10 °C until further use.

Free CNPSs used as blank (using 1 mg/mL) were prepared following the same method described above, with no addition of CVX to the C solution. Prior to the encapsulation process, the viscosity of all C prepared solutions was measured at room temperature using a Brookfield rotatory viscometer (Canary Wharf, London, UK).

### 2.3. Average Size and Zeta Potential Determinations

The average particle diameter, polydispersity index (PI), and zeta potential (ζ) were determined using a Zetasizer Zen Systems 3600 from Malvern Instruments Ltd. (Worcestershire, Malvern, UK). For Particle size and PI measurements, samples were diluted (1:50) with double-distilled water and were analyzed by dynamic light scattering (DLS) at a 90° scattering angle, for 180 s at 25 °C. ζ was analyzed by electrophoretic mobility (180 s/25 °C), where samples were diluted (1:50) with deionized water to avoid multiple scattering effects and placed in a folded capillary cell. All sizes, PI, and ζ measurements were made in triplicate, and the results are shown as mean ± standard error.

### 2.4. Morphology

Morphological studies were performed using a transmission electron microscope (TEM) JEOL JEM-1010 (Peabody, MA, USA) under 80–100 kV voltage acceleration. Samples were placed on a 300-mesh copper grid for observation. The shape and size were determined by analyzing at least 100 particles in different fields randomly using ImageJ^®^ (version 1.51) software.

### 2.5. CHIVAX 2.1 Viability and Encapsulation Efficiency

To evaluate the integrity of CVX after the encapsulation process, denaturing electrophoresis (SDS-PAGE) was implemented [12]. Briefly, CVX-loaded CNPs (100 mg) were incubated overnight in a Tween 20-enriched PBS (pH 7.4) at 37 °C with gentle shaking (≈100 rpm) to promote degradation of C. The released protein was collected by centrifugation at 2000 rpm/15 min/10 °C. Aliquots (5 mL) from supernatant were resuspended in SDS loading buffer and analyzed by SDS-PAGE. Determination of CVX recovered from CNPs recognition by anti-SARS-CoV2 antibodies was determined by Western blotting; the CVX electrophoresed on 10% SDS-PAGE gel was transferred onto a nitrocellulose membrane using a semi-dry transfer system. The system was assembled by placing the gel in contact with the nitrocellulose membrane in a semi-dry transfer chamber, moistening the system with transfer buffer (20 mM Tris-base, 120 mM glycine, and 20% methanol), and applying 15 V for 15 min. The transfer was corroborated using Ponceau-red staining on the nitrocellulose membranes, which was removed by washing with TBST solution (20 mM Tris-base, 150 mM NaCl, 0.1% Tween-20). Transferred nitrocellulose membranes were blocked by incubating them in agitation for 12 h with 5% nonfat dry milk diluted in TBST. Subsequently, the membranes were washed with TBST solution. Blocked membranes were incubated in agitation with serum obtained from pigs immunized with CVX for 1 h at 4 °C. The serum was diluted in a 1:15,000 TBST solution. Then, the membrane was washed with TBST and blocked again for 1 h. Then, the membrane was incubated in agitation with horseradish peroxidase (HRP)-conjugated anti-pig IgG antibodies (Bethyl, Montgomery, TX, USA) diluted in 1:25,000 TBST solution with 1% nonfat dry milk for 1 h at 4 °C.

Following incubation with the secondary antibody, the membrane was washed with TBST and placed in the dark tray of a ChemiDoc system (Bio-Rad, Hercules, CA, USA). The Amersham ECL Western blotting Detection Kit (Cytiva, Marlborough, MA, USA) was applied to the membrane, which was then imaged using the chemiluminescence program of the ChemiDoc system. A second colorimetric image was captured and merged with the selected chemiluminescence image.

To determine the encapsulation efficiency percentage (EE%), aliquots from the supernatant were analyzed for protein content by densitometry. The calibration curve was constructed using bovine serum albumin (BSA) at different concentrations (from 0.1 to 1 mg/mL), which were loaded onto a 10% acrylamide gel.

The Coomassie blue stain of the CVX-loaded NPs and BSA standards was photographed, showing each band on its corresponding molecular weight (Figure 1a).

The gel image (Figure 1) obtained from SDS-PAGE was analyzed using ImageJ software. The pixel density corresponding to each protein concentration was quantified and correlated with its respective pixel value (Table 1). The EE% was calculated using the following Equation (1):(1)$$EE\%=\frac{Amount\;of\;CVX\;encapsulated}{Amount\;of\;CVX\;before\;encapsulation\;process}\;\times \;100$$

### 2.6. Diffuse Reflectance of Infrared by Fourier Transforms (DRIFT)

Infrared spectra of C, CNPs, and CVX-CNPs were examined by DRIFT in a Spectrum GX spectrophotometer (Perkin Elmer, Waltham, MA, USA). To obtain samples with the minimum amount of humidity, all samples were placed in desiccators containing silica gel for at least 48 h at room temperature. Then, 2 mg of each sample was mixed with 98 mg of KBr (with no moisture content). The spectra were collected at 4 cm^−1^ resolution over the wavenumber range of 400–4000 cm^−1^. Each DRIFT spectrum is the result of 16 recorded scans, and the KBr spectrum was subtracted from each sample spectrum.

### 2.7. Mucin’s Adsorption of CNPs

The absorption of mucin on the surface of CNPs was carried out through modifications to the protocol established by He et al. [16] and Pawar and Jaganathan [17]. Briefly, equal volumes of CNPs (2 mg/mL) and mucin (0.5 mg/mL, in double-distilled water) solutions were mixed by magnetic stirring (300 rpm) for 90 min at 37 °C. This mixture was subjected to centrifugation (4000 rpm/2 min), and the supernatant was used to identify the concentration of free mucin by a colorimetric assay mediated by Periodic acid/Schiff (PAS), reported by Mantle and Allen [18]. The amount of mucin adsorbed on the surface of CNPs was calculated by subtracting the amount of free mucin from the amount of initial mucin used in this protocol (Equation (2))

Colorimetric assay was carried out at a wavelength of 500 nm using UV spectroscopy using the Periodic acid/Schiff (PAS) colorimetric method. The tests were carried out on at least 3 occasions, and results represent the average ± standard deviation.(2)$$Mucin\;binding\;efficiency\;(\%)=\frac{C_{O}-C_{S}}{C_{O}}\times 100$$

### 2.8. In Vitro Release Profiles

The in vitro release of CVX from CNPs was determined using a simulated nasal fluid (SNF) (Table 2) [15], enriched with 1% of Tween 20 (final concentration) to increase the solubility of the CVX protein and to maintain the sink conditions of the system. CVX release from CNPS was determined by incubating CVX-loaded CNPs in 50 mL of 1% Tween-enriched SNF (pH 6.3) with gentle shaking (≤50 rpm) at 37 °C. Nanoparticles were centrifuged at different periods of time for up to 6 h, and aliquots (5 mL) were taken and placed in sterile vials. After the aliquots were taken, 5 mL of fresh 1% Tween-SNF solution was added to maintain sink conditions. The released concentration of CVX was determined using the densitometry technique described previously. Cumulative release profiles were performed in triplicate, and the results are expressed as the average ± standard deviation at each point in time.

To predict the drug release profiles, two kinetic models: Higuchi and Korsmeyer–Peppas models were applied (Equations (3) and (4), respectively):(3)$$\frac{Mt}{M\infty }=k_{H}t^{1/2}\;$$(4)$$\frac{Mt}{M\infty }=kt^{n}$$

In this equation, “*Mt*” represents the amount of CVX released at time “*t*”. “*M∞*” represents the total amount of CVX released in “infinite time.” “*k_H_*” and “*kt*” are the release rate constants for the Higuchi and Korsmeyer–Peppas models, respectively. “*n*” represents the release exponent, which indicates the mechanism.

### 2.9. In Vivo Assays

All animal procedures in this study were performed following a protocol reviewed and approved by the Research and Ethics Committee of the Autonomous University of Queretaro (CEAIFI-159-2022-TL) and carried out by our Institutional Veterinarian. A total of 15 C57-BL/6J male mice, 9–10 weeks old, were purchased from the Animal Vivarium AUT-B-C-1120064 at the Autonomous University of Mexico, Campus Juriquilla (UNAM-INB). The animals were housed in standard caging and were provided with food and water (Conejina T, Purina, St. Louis, MO, USA) ad libitum with a 12:12 light–dark circadian cycle at 21 ± 2 °C and relative humidity of 60 ± 5%. The mice were then allowed to acclimate for two weeks (Figure 2a).

During the acclimation period, mice were trained in the intranasal delivery technique. This involved a period of handling and acclimating the mice to the modified scruff restraint, followed by a gradual introduction to intranasal administration with saline until they received the full dose volume. The modified scruff was achieved by using the non-dominant hand with the neck held parallel to the floor, while 0.9% NaCl was delivered with a P10 Pipettor using the dominant hand. This technique was selected due to its non-invasive nature, which facilitates the delivery of large molecules that cannot cross the blood–brain barrier directly to the Central Nervous System (CNS). Intranasal administration enables precise targeting of the CNS while minimizing systemic exposure, thereby reducing the likelihood of undesired systemic side effects [15]. Delivery from the nose to the CNS occurs within minutes along both the olfactory and trigeminal neural pathways via an extracellular route and does not require the drug to bind to any receptor or axonal transport [19].

The animals were randomly divided into five groups (*n* = 3) and two immunization schedules (0 and 21 days) were used. All animals included in this experiment were physically healthy at the start of the experiment and the 3Rs were considered to optimize the number of animals. After completion of the training period and full acclimatization of the mice to intranasal administration, mice were weighed and dosed with CVX and CNPs at various concentrations in a maximum total volume of 24 μL allowed by the dosing regimen (Figure 2b). The total volume administered was divided equally per nostril (4 sets of 3 µL drops per nostril) by using the P10 Pipettor. Throughout the duration of the experiment, our Institutional Veterinarian was tasked with the health monitoring of our mice and intervention in the event of any adverse effects; however, none were noted. After 31 days, the mice were humanely euthanized, and the turbinate, lungs, spleen, and liver were fixed in 10% formaldehyde for histopathological evaluation.

## 3. Results and Discussion

### 3.1. Physicochemical Properties of CNPs 

Since the physicochemical properties of micro- and nanoscale-controlled delivery systems directly influence their biological behavior [8], characteristics such as mean particle size, morphology, surface electrical charge, and cargo were evaluated.

Diverse parameters influence the particle size distribution of polymeric nanoparticles developed through the ionic gelation technique, including the concentration of wall material, the concentration of cross-linking agent, the polymer’s molecular weight, and the type and concentration of active compound to be encapsulated [10,20]. For this reason, the effect of chitosan concentration on viscosity, size distribution, and impact on encapsulation efficiency was evaluated and shown in Table 3. It was observed that the higher the concentration of chitosan (1, 2, and 5 mg/mL), the greater the viscosity of the solution (1.663 ± 0.002, 2.967 ± 0.005, and 6.012 ± 0.003 mPa·s, respectively). Likewise, as the concentration of chitosan increased, the average particle size was greater (38.5 ± 9.1, 92.3 ± 6.6, and 316.8 ± 8.5 nm, respectively). This indicates a clear association between wall material concentration, solution viscosity and the average particle size obtained. On the other hand, it was observed that by adding CVX, the average particle size increases significantly when compared to free nanoparticles (87.9 ± 10.6, 161.8 ± 11.9, and 542.5 ± 40.1 nm) but did not significantly affect the viscosity of the solution. These findings suggest a strong electrostatic influence of CVX on C, which directly impacts the particle size. Since the PDI values did not exceed 0.5, the synthesis process was well-controlled, resulting in good homogeneity in terms of particle size. Nanoparticle size is very important for our study, due to its impact on nanoparticle absorption by mucosal tissues. Smaller nanoparticles (20–120 nm) generally exhibit better mucus absorption mainly by their increased surface area, unlike larger nanoparticles (≥150 nm), which may struggle to penetrate mucus layers effectively. However, extremely small nanoparticles (≤10 nm) may be cleared too quickly from the mucus layer, which can represent a problem for the effective release and retention of antigens in the nasal mucosa [21].

Also, it was observed that the encapsulation efficiency (%) of CVX improved as the particle size increased. This may be because increasing the concentration of chitosan increases the viscosity, which augments the interaction and retention of the protein in the particles. These results are consistent with the results reported by other authors [22].

The surface electrical charge(ζ-potential) of CNPs had a strong positive charge (+38.8 mV), which decreased to +26.5 mV when CVX was incorporated into the nanoparticles produced with the same amount of chitosan (1 mg/mL). Similarly, as the concentration of encapsulated CVX increased, the zeta potential of the system decreased (to +11.7 and +4.6 mV, respectively), approaching a zwitterionic charge, indicating a strong electrostatic influence of the protein in the developed system. The surface electric charge is important because it indicates possible particle repulsion or aggregation in a diffusion medium. Particles with ζ values above −20 mV or +20 mV have greater electrostatic repulsion and therefore, less aggregation and greater colloidal stability [20]. These results are interesting, since a trend is noted: the higher the concentration of chitosan, the greater the amount of protein incorporated into the system, changing at the same time the physicochemical properties (mean particle size, zeta potential) of the entire system (as shown in Table 3). According to the observations of [23,24], the formation of a protein corona around chitosan nanoparticles can alter various physicochemical properties of the developed system (including porosity, particle size, zeta potential, stability), as well as its biological performance (including the magnitude of the immune response). To elucidate the formation of protein corona, various methodologies are implemented, such as UV-Vis spectroscopy analysis, infrared spectroscopy (FTIR) determinations, transmission electron microscopy (TEM) analysis, among others. Therefore, TEM and FTIR analysis are implemented in this study to determine the formation of CVX corona around chitosan nanoparticles.

TEM analysis confirmed the effective formation of chitosan nanoparticles, which were predominantly spherical or semi-spherical in shape. Parameters such as particle size, zeta potential, and encapsulation efficiency (EE%) are critical for selecting a system suitable for controlled-release applications in the biological field. According to the studies carried out by [25,26], the formation of protein corona around polymeric nanomaterials could be noticed as a faint halo around the particles when observed by TEM. Also, the presence of this protein corona could have an impact on the colloidal stability of the systems, causing agglomeration of the systems, which could be visible through TEM analysis. In our results, it was observed that as the concentration of chitosan increased, the amount of protein incorporated into the nanoparticles increased, causing an increment in particle size and system agglomeration (as observed in 3c). In the system prepared with 2 mg/mL of chitosan, it is observed that there is a significant agglomeration of the system, which could suggest the influence of the CVX corona, which should be greater due to having more protein incorporated into the system. On the other hand, in the system developed with the highest concentration of chitosan (5 mg/mL), it was noticed that the system increases significantly in particle size, there is not as much agglomeration, but electron-dense dots are observed distributed in the particles, which could explain the presence of a CVX corona on the surface of the nanoparticles. However, in the system formulated with the lowest concentration of chitosan (1 mg/mL), a well-defined morphology, good dispersion, and an ultra-thin halo around the particles were observed, which could suggest the presence of a small CVX corona around the particles.

These physicochemical characteristics could influence the antigenicity and immunogenicity of vaccines. Antibody production is specific to the adjuvant and antigen, and consequently, so is the resulting cellular immune response [13,27]. The duration and strength of the immune response depend on the type of adjuvants used, as well as on their homogeneous distribution and electric charge, which facilitate the up-take of antigen by antigen-presenting cells (APCs) [28]. Therefore, the CVX-CNPs systems prepared at a concentration of 1 mg/mL were selected for subsequent physicochemical characterization and biological performance assays, as they exhibited a particle size below 100 nm, a surface charge that ensures colloidal stability and prevents aggregation, and an encapsulated protein content greater than 1 mg/mL (Figure 3).

### 3.2. Immune Detection of CVX Recovered from CNPs

A crucial step in the development of nanoparticulated vaccine systems is to demonstrate that the physical and immunological integrity of the antigen remains intact after the biological encapsulation process. Therefore, a Western blot assay was performed to verify the immunodetection of the protein (Figure 4). The results showed that CVX extracted from the CNPs was successfully recognized by the pig serum antibodies, indicating that CVX retains both its structural and immunogenic integrity [12]. These findings suggest that the conditions established in this study are suitable for encapsulating this innovative chimeric protein, while preserving its capacity to elicit an immune response. Moreover, the intensity of the CVX band was more pronounced in nanoparticles formulated with higher amounts of chitosan, further supporting the encapsulation efficiencies reported in Table 3. Additionally, the chemiluminescent signal decreased proportionally with the concentration of CVX-loaded CNPs.

### 3.3. Spectroscopic Properties (FTIR Analyses) 

ATR-FTIR analyses were performed to characterize the chemical structures of both free and CVX-loaded CNPs (Figure 5a). The C spectrum shows characteristic peaks related to O–H stretching bonds (3324 cm^−1^), symmetric and asymmetric C–H bonds (2868 cm^−1^), C=O stretching bond in the N-acetyl group (amide I) (1653 cm^−1^), to the amino group (amide II) (1586 cm^−1^), to symmetrical deformations of CH_2_ and CH_3_ bonds (1427 cm^−1^, 1370 cm^−1^, and 1312 cm^−1^), to C-O-C bonding (1144 cm^−1^), to glycosidic bond (1067 cm^−1^), to o symmetric and anti-symmetric stretching vibrations in the PO_3_ group (1029 cm^−1^), and to the hydroxyl group (885 cm^−1^). Our results agree with studies reported by other authors [29,30]. In the spectrum of CVX-loaded CNPs. The same peaks related to the vibrational movements of the functional group bonds of C were observed; however, these peaks are slightly shifted, which may mean that the presence of the CVX protein causes these changes by having a direct influence on the bonds formed with its functional groups. Likewise, the presence of new peaks was observed, specifically in the frequency numbers of 2959 cm^−1^, 2911 cm^−1^, 1220 cm^−1^, 985 cm^−1^, 831 cm^−1^, and 716 cm^−1^. These new peaks indicate that the CVX protein is distributed or dispersed throughout the developed material (center, surface, etc.), which is characteristic when a matrix or nanosphere type system is obtained, and confirms our observations made through TEM analysis.

### 3.4. Mucin Adsorption

Nasal administration of antigens is a promising vaccination strategy due to its safety, efficacy, and high acceptability. Therefore, understanding the interaction between adjuvants and the nasal mucosa is a crucial step in the design of safe and effective intranasal vaccine candidates. Figure 5b shows the mucin-binding efficiency (%) of all the nanoparticle formulations evaluated. In all systems, the longer the interaction time, the greater the mucin adsorption on the surface of the CNPs.

The trend in mucin adsorption efficiency was as follows: CNPs > CVX-CNPs 1 (1 mg/mL chitosan) > CVX-CNPs 2 (2 mg/mL chitosan) > CVX-CNPs 3 (5 mg/mL). The outcomes of this study may be influenced by two parameters of the nanoparticles: zeta potential and average particle size. It has been determined that electrostatic interaction occurs between sialic acid residues of mucin, which possess a negative charge, and amino groups of C, which possess a positive charge [16]; therefore, those nanoparticles with the highest positive charge will present greater mucin adsorption (CNPs). Likewise, particles with smaller average sizes (CNPs, ≈39 nm) are those with the highest mucin adsorption efficiency. This phenomenon could be attributed to the fact that smaller particles have a larger specific surface area, which enhances their interaction with mucin and improves adsorption efficiency [31]. In the case of CVX-CNPs, the systems prepared with 1 mg/mL of chitosan (CVX-CNPs 1) exhibited the highest mucin adsorption efficiency over time. Interestingly, these nanoparticles were those that presented the smaller average size (≈88 nm) and the higher zeta potential (≈+27 mV) compared to the systems formulated with higher concentration (2 mg/mL and 5 mg/mL) of chitosan (≈162 nm/≈+12 mV; ≈543 nm/≈+5 mV, respectively). These results support the notion that average size and surface electrical charges are critical factors influencing mucin adsorption. Our findings suggest that controlling these physicochemical parameters is essential for designing more efficient intranasal vaccine candidates. The results of the physicochemical characterization study indicated that the particles formulated with 1 mg/mL of chitosan were optimal, as they exhibited an ideal average size (≈90 nm), a CVX load above the effective therapeutic concentration (1.316 mg/mL), and superior mucin retention on their surface (60–85% over time). So, in vitro release profiles and in vivo biocompatibility tests were conducted using these systems.

### 3.5. In Vitro Release Kinetics

The release kinetics of CVX protein from CNPs were evaluated by incubating the NPs in a pH 6.3 SNF. This experiment aimed to assess the potential of CNPs to deliver protein in an intranasal environment (Figure 5c). An immediate release of 89% of the CVX was observed within the first 120 min, followed by a sustained release of 90–91% of the protein within 400 min. The initial burst release could be attributed to the desorption of CVX adsorbed on or near the surface of the nanoparticles, while the subsequent slower release could be CVX encapsulated within the inner matrix of the CNPs. These results are characteristic of a matrix-type release system, which aligns with the results obtained in the ATR-FTIR analysis. Furthermore, our observations are consistent with previous studies evaluating protein release from CNPs in different diffusion media [23,24]. To explain and delve deeper into the CVX release profiles, the release data were fitted using the Higuchi and Korsmeyer–Peppas semi-empirical models (Table 4).

The results indicate that the protein release data fit both models well, with a better fit to the Higuchi model (R^2^ = 0.9951) than for the Korsmeyer–Peppas model (R^2^ = 0.977). According to the Higuchi model, spherical systems follow a pure Fickian diffusion when k_H_ values are ≤0.43. However, anomalous transport is shown when values are between 0.43 and 0.85. The k_H_ constant obtained for the CVX release kinetics was 0.1197, which suggests that CVX release may occur by diffusion. Within the Korsmeyer–Peppas model, the value of “*n*” indicates different behaviors. Values of *n* ≤ 0.45 suggest Fickian diffusion; values of 0.45 < *n* < 0.89 indicate anomalous (non-Fickian) transport; values of *n* = 0.89 imply Case II transport; and values of *n* > 0.89 are associated with Supercase II transport [32].

Our results showed that the value of *n* for CVX release data is 0.9092. This indicates a release phenomenon associated with Supercase II transport. Supercase II transport describes a drug release phenomenon that is not solely diffusion-based, but rather, is intrinsically linked to the polymer’s mechanical response to swelling. This generates stress and cracking in the polymer matrix, producing the subsequent rapid release of the encapsulated active ingredient. Based on these results, we suggest that the rapid release of CVX from CNPs may be due to a combination of mechanisms: diffusion of the protein associated with the surface of the nanoparticles and subsequent swelling and degradation of the CNPs in the medium (SNF). A study conducted by Wijayawardana et al. [33] has shown that, depending on the diffusion medium, materials made with chitosan exhibit two release phenomena related to Fickian diffusion and supercase II transport. These results are consistent with those obtained in this work. On the other hand, intranasal vaccine candidates can be formulated for immediate or sustained release depending on the desired immune response. Sustained release is optimal when antigen exposure must be maintained over time to promote a robust and prolonged mucosal and systemic immune response. However, immediate-release formulations can be beneficial for generating a rapid initial immune response or for delivering antigens quickly to bypass barriers [34,35]. For optimal administration of antigens within the nasal cavity, the sustained release could be the more desirable one, since it can release and maintain optimal concentration of the antigen in the nasal mucosa, which enhances the efficacy of the immune response [36]. However, for this type of release, it is necessary to evaluate the risk of mucosal toxicity due to prolonged exposure to the adjuvants/excipients [37]. According to our results, the CVS released from CNPs is immediate, which could be beneficial to generate a rapid activation of the immune response and to avoid a possible adverse effect on the nasal mucosa due to extreme exposure to the adjuvant material. However, it also provides an area of opportunity to improve certain features of our system, as well as to implement other types of wall materials that have been shown to have a more sustained release within the nasal cavity, such as PLGA [38].

### 3.6. In Vivo Assays

To evaluate the biosafety of our nanovaccine candidate, the body weight of the animals was monitored throughout the experimental period, from the initial administration (day 0) to humane euthanasia (day 31) (Figure 6). Weight monitoring served as a critical health parameter, enabling the detection of potential adverse effects or physiological stress induced by the CVX and CNPs [39]. Additionally, histological analyses were performed to correlate weight trends with tissue-level biosafety. The results showed no significant weight changes compared to the control group. A slight increase was observed in all groups over time (Figure 6), supporting the conclusion that the vaccine did not induce detectable systemic toxicity.

In this study, the liver, spleen, lungs, and nasal turbinates were evaluated both grossly and microscopically to identify potential toxicological lesions that could compromise vaccine safety (Figure 7). Mild to moderate congestion was observed in the lungs, spleen and liver across all groups, a finding previously reported in murine models and commonly attributed to the CO_2_ euthanasia method [40].

Histological analysis revealed no significant alterations in cellular or tissue architecture between the experimental groups. Additionally, in the spleen, white pulp hyperplasia and expansion of the marginal zones have been reported in cases of high-dose intranasal vaccine administration. However, these changes were not observed in this study. The primary target organs—the nasal turbinates and lungs- showed no toxicological alterations or relevant pathological findings (Figure 7a,b). In various studies involving intranasal vaccines, lesions such as lymphoplasmacytic infiltrates, respiratory epithelial hyperplasia, and in severe cases, focal necrosis have been observed as signs of local immune activation. In the lungs, mild interstitial pneumonitis, presence of perivascular inflammatory cells and lung edema have been found [41,42]. None of these pathological changes were detected following administration of the vaccine candidate.

## 4. Conclusions

The development of new intranasal vaccine candidates to combat diseases such as COVID-19 is closely linked to the use of adjuvant materials capable of carrying, releasing, and effectively retaining antigens within the nasal mucosa, while avoiding adverse effects on the host. In this work, chitosan was employed as a delivery platform due to its unique properties, including high biocompatibility, biodegradability, and its ability to stimulate the immune response.

Our findings demonstrate that the experimental conditions established in this work are optimal for the encapsulation of CHIVAX 2.1, preserving its physical integrity and its previously reported immunogenic properties. Furthermore, the concentration of chitosan was shown to play a crucial role in determining both the average nanoparticle size and the amount of protein encapsulated, with higher chitosan concentrations yielding larger particles and greater protein loading. Although the particles produced with the lowest chitosan concentration encapsulated a smaller amount of protein, the resulting antigen concentration remained above the effective immunogenic threshold for CHIVAX 2.1. Notably, these smaller particles also exhibited superior mucin retention, likely due to their higher surface area. Importantly, the present study demonstrated that the CVX-loaded CNPs vaccine candidate does not induce adverse effects in the respiratory tract, liver, and spleen of experimental animals, indicating a high degree of biocompatibility. Nevertheless, further studies are necessary to ascertain the vaccine efficacy, which is crucial for determining not only its biocompatibility but also its capacity to elicit a protective immune response against severe acute respiratory syndrome (SARS-CoV-2). Finally, this study represents the first report of an attempt to encapsulate a chimeric protein created and patented by our research group (CVX), with polymeric nanoparticles specifically designed for intranasal administration, with the goal of providing protection against COVID-19 infection and its associated complications.

## Figures and Tables

**Figure 1 biomedicines-13-02453-f001:**
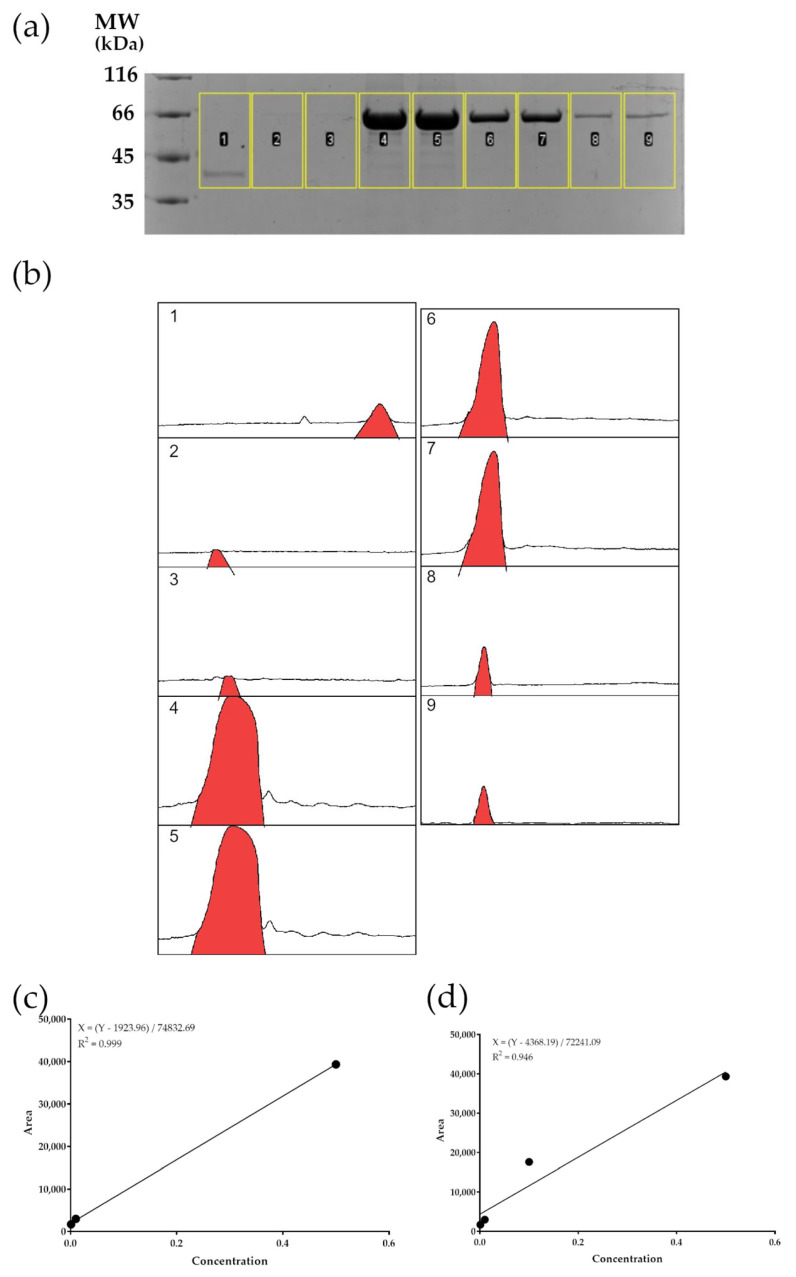
Generation of a standard curve to calculate the CVX loaded into CNPs. (**a**) SDS-PAGE of CVX recombinant protein extracted from CNPs BSA samples with known concentration. (**b**) Densitometry curves generated by ImageJ. The panels correspond to each sample as follows: 1: CVX-loaded CNPs; 2–3: BSA at 0.001 mg/mL; 4–5: 0.5 mg/mL; 6-7: 0.1 mg/mL; 8–9: 0.01 mg/mL. (**c**) Standard curve generated using all points. This regression model achieved an R^2^ score of 0.946. The regression line equation is shown in the upper-left portion of the graph. (**d**) Standard curve generated using all points except those corresponding to 0.1 mg/mL. This regression model achieved an R^2^ score of 0.999. The regression line equation is shown in the upper-left portion of the graph.

**Figure 2 biomedicines-13-02453-f002:**
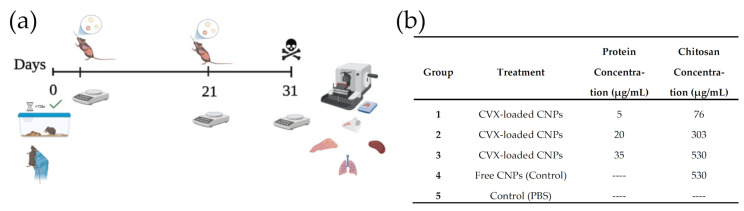
Design of an in vivo assay. (**a**) Administration scheme: two intranasal doses are given on days 0 and 21. (**b**) Treatments administered to the different groups of experimental animals: Groups 1 to 3 received treatment based on the complete CVX-CNPs system, increasing the concentration of encapsulated recombinant protein; Group 4 received treatment based on free CNPs; Group 5 (Control) received PBS.

**Figure 3 biomedicines-13-02453-f003:**
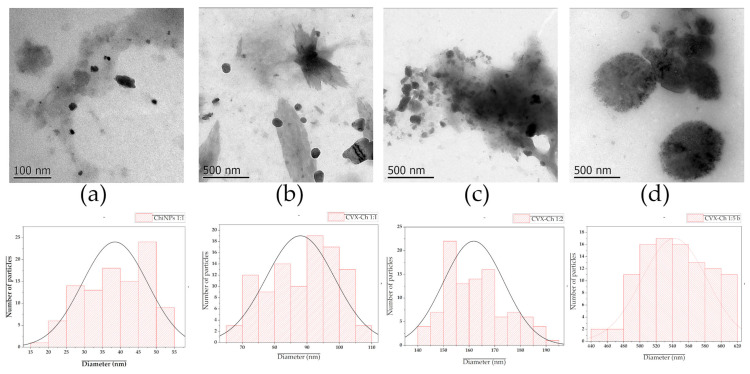
Morphology analysis by transmission electron microscopy (TEM) of chitosan-free nanoparticles (**a**) and CHIVAX 2.1 (CVX)-loaded chitosan nanoparticles, implementing different concentrations of chitosan, 1 mg/mL (**b**), 2 mg/mL (**c**), and 5 mg/mL (**d**). Histograms indicate the size distribution of each sample, after analyzing the TEM micrographs using ImageJ^®^ (version 1.54) software.

**Figure 4 biomedicines-13-02453-f004:**
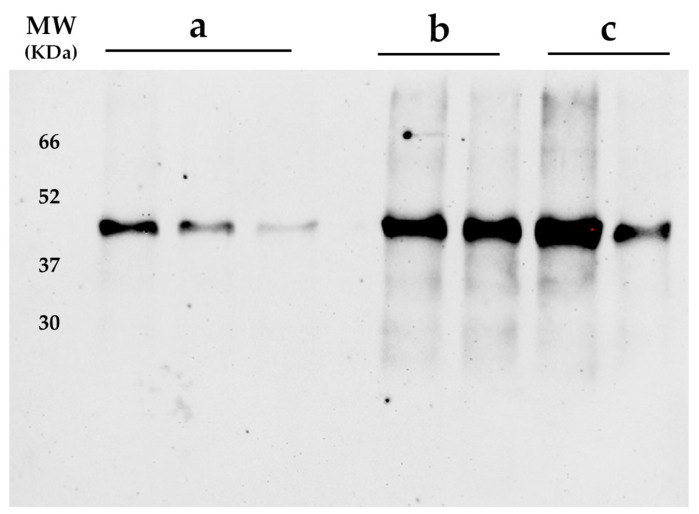
Immunodetection of CVX extracted from CNPs formulated at different concentrations of C: (**a**) 1 mg of chitosan, (**b**) 2 mg of chitosan, and (**c**) 5 mg of chitosan. The CVX-loaded CNPs were successfully detected by the anti-CHIVAX 2.1 antibodies in an immunized pig. The signal decreased as the concentration of the nanoparticles decreased. (MW) Molecular weight marker (Applied Biological Materials Inc (Richmond, BC, Canada). Opti-Protein Ultra Marker).

**Figure 5 biomedicines-13-02453-f005:**
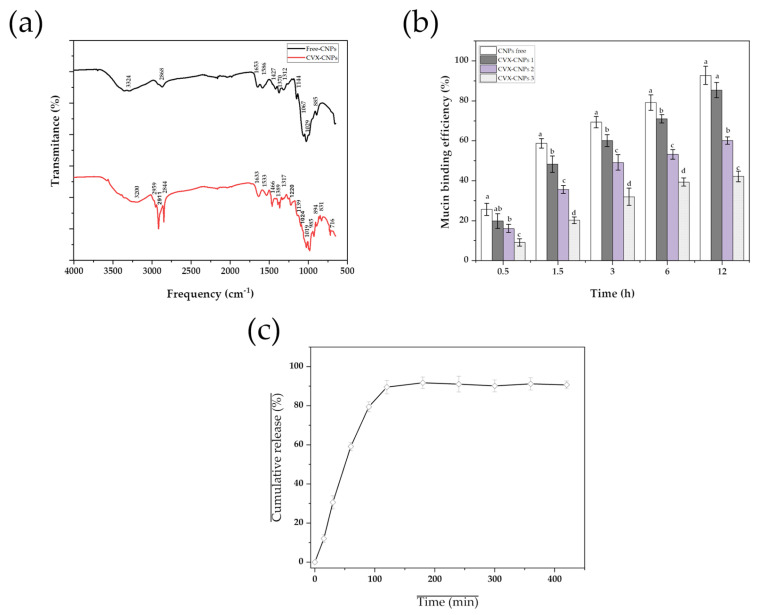
(**a**) FTIR spectra of free- and CHIVAX 2.1-loaded chitosan nanoparticles. (**b**) Mucin binding efficiency (%) of chitosan nanoparticles prepared with different concentrations of chitosan: 1 mg/mL (Free and CVX-CNPs 1), 2 mg/mL (CVX-CNPs 2), and 5 mg/mL (CVX-CNPs 3) as a function of time. All results are expressed as mean ± SE of three repetitions with three replicates. Different letters indicate a significant difference (*p* < 0.05) between samples at each time by Dunnet’s test. Statistical analyses were performed using Origin software 8.0. (**c**) In vitro controlled release profiles of CHIVAX 2.1 from chitosan nanoparticles in a pH 6.3 simulated nasal fluid.

**Figure 6 biomedicines-13-02453-f006:**
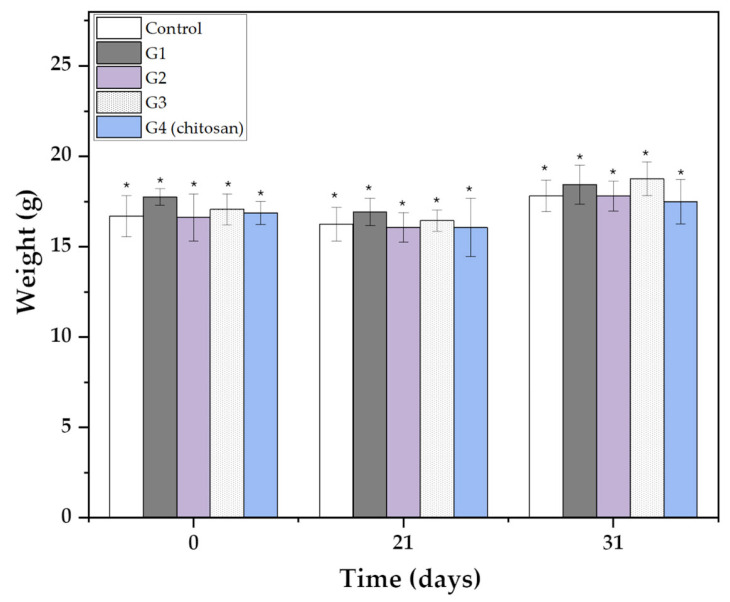
Weight of animals over time. The results are expressed as average ± standard deviation (SD). An unpaired *t*-test with Dunnet’s comparison with the control group for different postvaccine times (days 21 and 31) was performed. * Indicate no statistical difference in comparison with control by Dunnet’s test (*p* < 0.05). Statistical analyses were performed using Origin software 8.0.

**Figure 7 biomedicines-13-02453-f007:**
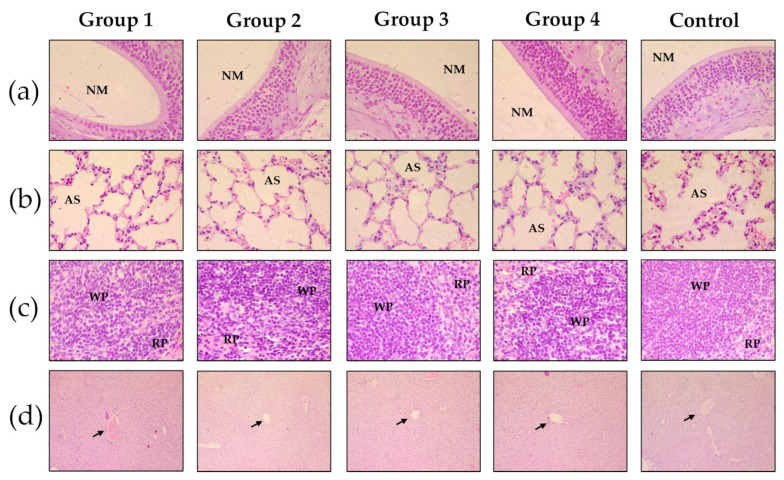
Histological analysis of organs from mice administered with CVX-CNPs at 31 days, stained with H&E. group 1 (CVX plus CNPs 5 μg); group 2 (CVX plus CNPs 20 μg); group 3 (CVX plus CNPs 35 μg); group 4 (CNPs 530 μg); control group. (**a**) Histological sections of nasal turbinates at ×63 magnification; NM: nasal meatus. (**b**) Histological sections of lung tissue at ×63; AS: alveolar space. (**c**) Histological images of spleen tissue at ×63; PB: white pulp; PR: red pulp. (**d**) Histological sections of liver tissue at ×10 magnification; Arrow: central vein. Slight congestion was present in all groups, but no apparent toxicological changes were observed.

**Table 1 biomedicines-13-02453-t001:** Area of the CVX-loaded CNPs and BSA standards bands obtained from the pixel density quantification using the image J gel-analyzer tool.

Sample	Area
CVX-loaded CNPs	5100.25
BSA [0, 5]	39,663.02
BSA [0, 5]	39,006.55
BSA [0, 1]	17,787.853
BSA [0, 1]	17,413.489
BSA [0, 01]	3315.82
BSA [0, 01]	2641.60
BSA [0, 001]	1558.11
BSA [0, 001]	1837.70

**Table 2 biomedicines-13-02453-t002:** Composition of simulated nasal fluids (SNF) at pH 6.3 [15].

Component	Concentration
NaCl	7.45 g/L
KCl	1.29 g/L
CaCl_2_	0.32 g/L
Double distilled water	q.s.

**Table 3 biomedicines-13-02453-t003:** Physicochemical properties of free- and CVX-loaded CNPs, and their encapsulation efficiency.

System	C Concentration (mg/mL)	Viscosity (mPa·s)	Average Size (nm)	Polydispersity Index (PDI)	ζ-Potential (mV)	Encapsulation Efficiency (%)
CNPs	1	1.663 ± 0.002 ^a^	38.5 ± 9.1 ^a^	0.22 ± 0.05 ^a^	+38.8 ± 1.2 ^a^	---
	2	2.967 ± 0. 005 ^b^	92.3 ± 6.6 ^b^	0.28 ± 0.04 ^ab^	+40.4 ± 2.5 ^a^	---
	5	6.012 ± 0.003 ^c^	316.8 ± 8.5 ^c^	0.30 ± 0.07 ^ab^	+46.5 ± 5.1 ^a^	---
CVX-CNPs	1	1.671 ± 0.005 ^a^	87.9 ± 10.6 ^d^	0.29 ± 0.08 ^ab^	+26.5 ± 2.6 ^b^	65.4 ± 3.8 ^a^
	2	2.976 ± 0.008 ^b^	161.8 ± 11.9 ^e^	0.36 ± 0.11 ^ab^	+11.7 ± 1.8 ^c^	84.8 ± 0.9 ^b^
	5	6.007 ± 0.004 ^c^	542.5 ± 40.1 ^f^	0.41 ± 0.09 ^b^	+4.6 ± 1.2 ^d^	92.2 ± 1.5 ^c^

^a,b,c,d,e,f^ = Different letters indicate significant differences (*p* < 0.05) between systems developed with the same chitosan concentrations.

**Table 4 biomedicines-13-02453-t004:** Parameters of Higuchi and Korsmeyer–Peppas models of CVX release rate from CNPs.

Higuchi		Korsmeyer–Peppas	
k_H_	R^2^	*n*	R^2^
0.1197	0.9951	0.9092	0.977

## Data Availability

The study included original contributions which are outlined in the article. If you have any more questions, please get in touch with the authors.

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
