# Peer review of "CHIVAX 2.1-Loaded Chitosan Nanoparticles as Intranasal Vaccine Candidates for COVID-19: Development and Murine Safety Assessment"

_biomedicines, 2025, doi:10.3390/biomedicines13102453_

Round 1
Reviewer 1 Report
Comments and Suggestions for Authors
This manuscript addresses the highly relevant challenge of improving COVID-19 vaccine delivery by exploring a novel intranasal formulation. It details the development and in vivo safety assessment of CHIVAX 2.1-loaded chitosan nanoparticles, presenting promising physicochemical properties and a favorable safety profile. The manuscript presents a valuable investigation into a novel intranasal COVID-19 vaccine candidate. The work addresses a highly relevant topic by aiming to overcome limitations of conventional vaccines. While the study provides promising initial data, several critical questions regarding the characterization and mechanism of the nanoparticle system need to be addressed to strengthen the conclusions and enhance the scientific rigor of the manuscript.
- While encapsulation efficiency is discussed, a more critical set of parameters, especially for drug delivery systems, relates to the loading capacity of the particles and associated characteristics. Beyond the fundamental aspect of size distribution, crucial properties such as Surface Area, Pore Volume, and Pore Size are paramount. Surface area, in particular, is noted by the authors (lines 321-322) to influence penetration efficacy, highlighting its importance. It would be valuable to investigate these parameters. Furthermore, it is plausible that the loading of the vaccine itself could influence the surface area and pore characteristics of the nanoparticles. Therefore I would recommend exploring this potential interplay.
- The mechanism of vaccine loading into the chitosan nanoparticles is not entirely clear. Is it possible that the vaccine electrostatically adheres to the particle surface, forming a dense shell? Clarifying the pore size distribution of the chitosan nanoparticles could offer significant insights into this question. A more detailed elucidation of the loading mechanism would greatly enhance the understanding of the system's design and function. The significance of encapsulation efficiency is also limited without presenting the mass ratio of chitosan to vaccine and the actual loading capacity.
- Furthermore, the conclusion in lines 438-440, "This phenomenon could be attributed to the fact that smaller particles have a larger specific surface area, which enhances their interaction with mucin and improves adsorption efficiency," warrants further discussion. Given that a reduction in zeta potential was also observed concurrently with changes in size (Table 3 shows a significant increase in mean carrier size after vaccine loading), this collective data may suggest a modification in the particles' surface chemistry. This again points to the need for a more comprehensive understanding of the particle's structure and surface properties after vaccine loading in addition to already provided FTIR data.
- Table 3 indicates a notable increase in the average size of the carriers after vaccine loading. This observation suggests that the vaccine may be forming a dense, thick layer or shell around the chitosan nanoparticles. If this is the case, it becomes critical to discuss how the mucoadhesive properties of chitosan are influenced by this vaccine coating, and how this impacts the vaccine's kinetics upon intranasal administration. Additional structural characterization techniques beyond FTIR could provide invaluable information to support these discussions.
- Typographical Error: There appears to be a typographical error on line 464.
- The axes and labels in Figure 3 are currently not clearly legible. I would recommend enhancing their clarity and size for better interpretation of the presented data.
- The article states that up to 89% of release occurs within the first 20 minutes, whereas Figure 7 suggests that approximately 90% release is achieved only after about 120 minutes. Please clarify whether Figure 7 represents the total release, or if the initial burst release phase is accounted for differently. Furthermore, it would be highly beneficial to discuss what type of release profile (e.g., immediate, sustained, biphasic with specific kinetics) is considered most desirable or effective for intranasal vaccine delivery systems in terms of optimizing immunization efficacy.
The authors have clearly conducted an extensive amount of work, and the research presents highly promising results for a relevant therapeutic area. The observations and questions raised in this review are primarily intended to stimulate further discussion, clarify mechanistic insights, and enhance the overall accessibility and scientific depth for the reader. The general quality of the conducted research is high. After addressing these points and making revisions, the manuscript will be suitable for publication.
Author Response
Thank you for your comments, which helped us improve the quality of this manuscript. Below are the answers to your questions and comments:
- While encapsulation efficiency is discussed, a more critical set of parameters, especially for drug delivery systems, relates to the loading capacity of the particles and associated characteristics. Beyond the fundamental aspect of size distribution, crucial properties such as Surface Area, Pore Volume, and Pore Size are paramount. Surface area, in particular, is noted by the authors (lines 321-322) to influence penetration efficacy, highlighting its importance. It would be valuable to investigate these parameters. Furthermore, it is plausible that the loading of the vaccine itself could influence the surface area and pore characteristics of the nanoparticles. Therefore I would recommend exploring this potential interplay.
We would like to thank the reviewer´s suggestion. The main objective of the work was to determine whether the CHIVAX 2.1 protein was capable of being incorporated into chitosan nanoparticles and whether this process affected their immunogenic properties. We were able to answer this first objective because of the characterization studies we conducted. Some of these methods include looking at the size of the particles, how charged the particles are, and taking pictures of the particles under a microscope. They also measure how well the particles can hold their cargo and how quickly they release it. (We attach DOI of such studies:
https://doi.org/10.1016/j.ijbiomac.2024.133964; https://doi.org/10.1016/j.micpath.2019.103600;
https://doi.org/10.1038/s41598-023-34448-6;
https://doi.org/10.1007/s13204-018-0779-8; https://doi.org/10.3390/pharmaceutics17010132
In our work, we not only analyze these physicochemical properties together, but we also determine an essential step that many studies do not evaluate. This step is determining the immunogenicity of the chimeric protein after the encapsulation process. We consider that the work we have done is supported by the methods we used and the results we obtained. These results are like those of many other studies that analyzed the same or fewer properties than in our work (between 2018 and 2025).
However, we think the reviewer's suggestion is very good, so we will look for equipment that can help us figure out the porosity and surface properties of these nanoparticles and other systems we are developing. These analyses will help us understand the encapsulation process better. We will learn how an active ingredient that is encapsulated plays an important role in certain properties of the total system. Some of these properties are the porosity, size distribution, encapsulation efficiency, surface properties, and release kinetics.
- The mechanism of vaccine loading into the chitosan nanoparticles is not entirely clear. Is it possible that the vaccine electrostatically adheres to the particle surface, forming a dense shell? Clarifying the pore size distribution of the chitosan nanoparticles could offer significant insights into this question. A more detailed elucidation of the loading mechanism would greatly enhance the understanding of the system's design and function. The significance of encapsulation efficiency is also limited without presenting the mass ratio of chitosan to vaccine and the actual loading capacity.
We appreciate the reviewer's comments and suggestion. Indeed, a direct porosity study would help us understand where the protein is in the nanoparticulate system. However, our working group does not have the specialized equipment needed to measure porosity and the adsorption of molecules on the surface of the particles. Therefore, it's difficult to make this decision right now. However, we are working to team up with a working group that can help us measure these surface properties and provide a better understanding of the encapsulation, characterization, and controlled release processes for this type of system and others we will carry out in the future. Our work on FTIR analysis also shows that there are absorption bands in the system that are not found in the system without protein. However, these absorption bands are very scarce, so it can be assumed that very little of the protein is stuck to the surface. These results show that the active ingredient is found inside the particle, near the surface, or on the surface, at a specific concentration. As we wrote in our manuscript, some studies have reported something similar using FTIR analysis:
https://doi.org/10.1016/j.carbpol.2014.05.005; https://doi.org/10.1080/00914037.2011.617334; https://doi.org/10.3390/pharmaceutics11020072
- Furthermore, the conclusion in lines 438-440, "This phenomenon could be attributed to the fact that smaller particles have a larger specific surface area, which enhances their interaction with mucin and improves adsorption efficiency," warrants further discussion. Given that a reduction in zeta potential was also observed concurrently with changes in size (Table 3 shows a significant increase in mean carrier size after vaccine loading), this collective data may suggest a modification in the particles' surface chemistry. This again points to the need for a more comprehensive understanding of the particle's structure and surface properties after vaccine loading in addition to already provided FTIR data.
The reviewer is right. However, as we mentioned earlier, our working group doesn't have the specialized equipment needed to measure surface properties and how compounds are absorbed in our nanoparticle systems. We have used FTIR spectroscopy to establish an analysis that can give us a close idea of what happens in the system we have developed. However, we are working with a research group to study this issue and better understand how the material and active ingredient affect the size, stability, and release of the particles. We really appreciate this information.
- Table 3 indicates a notable increase in the average size of the carriers after vaccine loading. This observation suggests that the vaccine may be forming a dense, thick layer or shell around the chitosan nanoparticles. If this is the case, it becomes critical to discuss how the mucoadhesive properties of chitosan are influenced by this vaccine coating, and how this impacts the vaccine's kinetics upon intranasal administration. Additional structural characterization techniques beyond FTIR could provide invaluable information to support these discussions.
We appreciate the reviewer's comment, since some studies have shown that when there is a greater amount of a protein, it generates a crown above the polymeric particle. In our study, we can see changes in the wavenumber and some new absorption bands. This suggests that part of the protein could be located near or on the surface of the nanoparticle system. However, the spectrum of CVX-CNPs is like the empty CNPs system. If a protein corona were to be formed, a completely different set of infrared (IR) light absorption peaks would be expected than that of the free system, since the IR radiation would come into contact exclusively with the protein corona and not with the polymer particle. This is something that is not noticeable in our Fourier-transform infrared (FTIR) spectrum. We are interested in this idea. We want to do this type of study in the future. We will look for special equipment. This equipment will help us know more about how biomolecules are spread in a nanoparticulate system.
- Typographical Error: There appears to be a typographical error on line 464.
Thank you for your observation, the correction was done in line 458 (464 before).
- The axes and labels in Figure 3 are currently not clearly legible. I would recommend enhancing their clarity and size for better interpretation of the presented data.
We appreciate the reviewer's suggestion. Image 3 has been improved for better understanding.
- The article states that up to 89% of release occurs within the first 20 minutes, whereas Figure 7 suggests that approximately 90% release is achieved only after about 120 minutes. Please clarify whether Figure 7 represents the total release, or if the initial burst release phase is accounted for differently.
We appreciate the reviewer's comments on this issue. In the text, we changed 20 minutes to 120 minutes, so there's a correlation with Figure 7. The explanation is included in lines 453-455.
7b. Furthermore, it would be highly beneficial to discuss what type of release profile (e.g., immediate, sustained, biphasic with specific kinetics) is considered most desirable or effective for intranasal vaccine delivery systems in terms of optimizing immunization efficacy.
The authors would like to thank the reviewer's suggestion. We fitted the CVX release data to two semi-empirical models, the Higuchi and Korsmeyer-Peppas kinetic models, to better understand and explain protein release phenomena. Similarly, we discuss whether CVX release from CNPs is optimal for a formulation intended for intranasal administration.
Therefore, the methodology (lines 250-260) and results (lines 451-501) sections on controlled release have been modified and improved for clarity. The paragraph has been revised to include Table 4, which contains the calculated constant values for both kinetic models.
Finally, we would like to thank for considering our work for publication in your prestigious journal.

Reviewer 2 Report
Comments and Suggestions for Authors
The manuscript describes the development and characterization of CHIVAX 2.1 (CVX), a chimeric SARS-CoV-2 protein, encapsulated within chitosan nanoparticles (CNPs) for intranasal delivery as a COVID-19 vaccine candidate. The study explores nanoparticle synthesis, physicochemical properties (size, zeta potential, morphology, encapsulation efficiency), mucin adsorption, in vitro release kinetics in simulated nasal fluid, and in vivo murine safety using histopathology and body weight monitoring. The results indicate high encapsulation efficiency, favorable mucin adhesion, rapid protein release, and an absence of adverse effects in treated mice. I recommend acceptance of the article with some minor correction.
- Consistency is needed in the use of abbreviations. For example, “CNPs” and “CNP” are both used. Please “µg” instead of “ug,” and ensure all units are consistent. Also, the scientific tone can be improved by avoiding overly conversational phrases.
- What was the rationale behind chitosan nanoparticles? Please discuss in introduction.
- The observed burst release (≈89% within 20 min) may enhance initial mucosal exposure but risks rapid clearance or degradation before sufficient immune activation. Are there any concerns?
- Please include scale bars on your TEM images
Author Response
Thank you for your comments, which helped us improve the quality of this manuscript. Below are the answers to your questions and comments:
- Consistency is needed in the use of abbreviations. For example, “CNPs” and “CNP” are both used. Please “µg” instead of “ug,” and ensure all units are consistent. Also, the scientific tone can be improved by avoiding overly conversational phrases.
Thank you for your observation, the correction was done in line 296. We also check the abbreviations and make sure they're mixed together correctly.
- What was the rationale behind chitosan nanoparticles? Please discuss in introduction.
Thank you for your question. The explanation is included in lines 79-80.
- The observed burst release (≈89% within 20 min) may enhance initial mucosal exposure but risks rapid clearance or degradation before sufficient immune activation. Are there any concerns?
We appreciate your commentary. A better explanation was included in lines 451-501.
- Please include scale bars on your TEM images
Thank you for your observation, figure 3 has scale bars in each image.
Finally, we would like to thank for considering our work for publication in your prestigious journal.

Reviewer 3 Report
Comments and Suggestions for Authors A very interesting manuscript regarding the possibility of creating a new vaccine formulation was reviewed. All experiments were conducted correctly, and the conclusions drawn are based on experimental data. My only comment concerns the quality of the figures. For example, figure 2 requires improvement. Furthermore, it seems reasonable to place figures 5, 6, 7, and 8 in one or two panels, which would allow for standardized size and make them more readable.Author Response
Comments and Suggestions for Authors
A very interesting manuscript regarding the possibility of creating a new vaccine formulation was reviewed. All experiments were conducted correctly, and the conclusions drawn are based on experimental data. My only comment concerns the quality of the figures. For example, figure 2 requires improvement. Furthermore, it seems reasonable to place figures 5, 6, 7, and 8 in one or two panels, which would allow for standardized size and make them more readable.
The authors appreciate the reviewer's suggestion. Due to figures must meet an optimal quality standard for the prestige of the journal Biomedicines, we believe that creating two panels for figures 5, 6, 7, and 8 could impact their quality and clarity. Therefore, we consider these figures should remain in their original format. However, we appreciate the reviewer's comment.
Finally, we would like to thank for considering our work for publication in your prestigious journal.
